# Visualization of Prediction Methods for Wildfire Modeling Using CiteSpace: A Bibliometric Analysis

Mengya Pan [†] and Shuo Zhang [†,*]

School of Information Management, Nanjing University, Nanjing 210023, China; pmy@smail.nju.edu.cn
* Correspondence: zhangshuo@smail.nju.edu.cn
† These authors contributed equally to this work.

**Abstract:** Wildfire is a growing concern worldwide with significant impacts on human lives and the environment. This study aimed to provide an overview of the current trends and research gaps in wildfire prediction by conducting a bibliometric analysis of papers in the Web of Science and Scopus databases. CiteSpace was employed to analyze the co-occurrence of keywords, identify clusters, and detect emerging trends. The results showed that the most frequently occurring keywords were "wildfire", "prediction", and "model" and the top three clusters were related to "air quality", "history", and "validation". The analysis of emerging trends revealed a focus on vegetation, precipitation, land use, trends, and the random forest algorithm. The study contributes to a better understanding of the research trends and gaps in wildfire prediction and provides recommendations for future research, such as incorporating new data sources and using advanced techniques.

**Keywords:** wildfire prediction; Web of Science; Scopus; CiteSpace; research trends

## 1. Introduction

Wildfires, defined as uncontrolled fires that occur in vegetation, forests, or other wildlands, are natural disasters that have significant impacts on the environment, economy, and society [1]. The consequences of wildfires are wide-ranging and long-lasting [2]. Environmental impacts include the loss of biodiversity, soil erosion, and air pollution. Economic impacts encompass damage to property and infrastructure and loss of timber and other resources, as well as the expenses associated with fire suppression efforts [3]. Social impacts involve the displacement of communities, adverse health effects, and psychological stress. The frequency and severity of wildfires vary across regions and are influenced by factors such as climate, vegetation type, and human activities [4]. The increasing frequency and severity of wildfires in recent years have emphasized the necessity of effective wildfire prediction models. Predictive models play a crucial role in mitigating damage and enhancing response strategies by enabling early detection of and rapid response to wildfires [5].

Wildfire prediction has been a subject of research for several decades, and recent years have witnessed significant advancements due to the emergence of new technologies and data sources. The utilization of remote sensing techniques [6], geographic information systems (GISs) [7], and machine learning algorithms [8] has facilitated the development of more accurate and efficient wildfire prediction models [9]. Furthermore, the availability of large datasets from diverse sources, such as satellite imagery [10] and the Internet of Things [11], has enabled the integration of multiple data sources and the creation of more comprehensive wildfire prediction systems.

Nevertheless, despite these notable advancements, the field of wildfire prediction still has numerous challenges and research gaps [12]. The complexity and unpredictability of wildfires, the heterogeneity of data sources, and the absence of standardization for methods and data formats present significant hurdles in the development of effective wildfire forecast models [13]. Consequently, there is an urgent requirement for research

endeavors aimed at identifying and tackling these challenges that would ultimately lead to the development of more precise and dependable wildfire prediction methods.

The field of wildfire prediction has witnessed a surge in studies due to the increasing popularity of mathematical statistics and artificial intelligence algorithms. Conducting a comprehensive review of these studies is crucial for understanding the research patterns in this domain. Bibliometric analysis [14], a method that utilizes mathematical and statistical tools, has been widely employed to analyze publications, citations, and journals across various knowledge areas. By developing bibliometric maps [15], researchers can gain a deeper understanding of their specific field, and scientific methodologies can be employed to track the evolution of research in wildfire prediction. Bibliometric analysis is a suitable approach for highlighting the primary findings of the literature and identifying key knowledge gaps. Additionally, text-mining techniques [16] can be applied to identify patterns within the scientific literature, facilitating an analysis of thematic, methodological, and conceptual trends over time. This aids in knowledge perception and the structuring of both developing and established scientific fields.

The specific research objective of this study was to investigate wildfire forecasting methods over the past 48 years and analyze recent research trends and advancements. The aim was to conduct a comprehensive bibliometric analysis of the literature pertaining to wildfire prediction, with a focus on identifying key research themes, influential authors, research hotspots, and research gaps. The study aimed to offer insights into the evolution of the field of wildfire prediction and propose potential directions for future research. The scope of the study was limited to the Web of Science and Scopus databases, covering the period from 1974 to 2022. Additionally, CiteSpace was utilized as a bibliometric analysis tool to unveil patterns for journals, terminology, countries, and author networks. The main work can be summarized as follows:

(1) ComprehensiveAnalysis: This study provides a comprehensive analysis of wildfire prediction research by examining various topics, such as publication trends, journal distributions, author networks, institutional networks, national networks, and keyword co-occurrences, as well as undertaking timeline analysis and emergent word analysis. This comprehensive approach offers a holistic understanding of the research landscape in wildfire prediction;

(2) Identification of research themes: Through the analysis of keywords and clustering, this study identifies key research themes in wildfire prediction. It highlights the dominant areas of research, including climate change, fire behavior, computer modeling, and smoke. This identification of research themes helps to delineate the major focus areas in the field;

(3) Mapping collaboration networks: We analyzed author networks, institutional networks, and national networks to uncover collaboration patterns and identify influential authors, institutions, and countries in wildfire prediction research. This information provides insights into collaborative relationships and knowledge dissemination within the research community;

(4) Visualization of research patterns: The use of visualization techniques, such as network diagrams and timeline analysis, helps to visualize and understand the evolution of research hotspots over time. It allows researchers to observe the dynamics of research themes, emerging trends, and the interconnections between different topics.

Overall, this study contributes to the existing knowledge on wildfire prediction by providing a comprehensive analysis of the research landscape. It offers valuable insights for researchers, practitioners, and policymakers and highlights the importance of continued research and collaboration in addressing the challenges posed by wildfires. The rest of this paper is organized as follows:

In Section 2, we present the materials and methods used for the bibliometric analysis. Section 3 provides the results of the analysis, including general information, journal distribution analysis, network analysis, and keyword analysis. Section 4 discusses the

findings in detail. Finally, Section 5 concludes the paper and highlights the contributions and research gaps identified in this study.

## 2. Materials and Methods

### 2.1. Data Source

The data utilized in this study were sourced from the Web of Science and Scopus databases [17]. Our search of the literature revealed that the earliest paper on wildfire prediction was published in 1974; thus, our data covered the period from 1 January 1974, to 31 December 2022. The search query employed was "Topic = (wildfire or forest fire or wildland fire) AND (prediction or forecast)", which yielded a total of 4524 publications. The search was conducted on 19 February 2023, and the search results were exported in both plain text and as full records with cited references using RefWorks.

### 2.2. Literature Selection Criteria

(1) Inclusion Criteria:The literature selected for this study focused on wildfire prediction research, encompassing topics such as wildfire danger rating [18], fire spread prediction, and post-disaster assessment.
(2) Exclusion Criteria:

- Articles unrelated to the topic, such as achievements, conference papers, patents, advertisements, popular science articles, etc;
- Non-original research, such as systematic reviews, meta-analyses, and reviews of wildfire prediction research;
- Articles with incomplete information, such as author, year, keywords, etc;
- Duplicate or withdrawn publications.

After applying the screening criteria, a total of 4305 papers were included as the sample data for this study.

### 2.3. Analysis Tools

CiteSpace, a citation visualization analysis software package, was utilized in this study to analyze the potential knowledge contained in the scientific literature. CiteSpace is specifically designed for scientific metrics and data visualization, aiming to uncover dynamic development patterns within disciplines and identify research frontiers. The version used in this study was CiteSpace 6.2.R3 (64-bit) Advanced, developed by Dr. Chaomei Chen and their team [19,20]. CiteSpace was used to process citations from the Web of Science and Scopus databases and generate interactive visualizations that depict structural and temporal patterns and trends in the field of wildfire prediction. It facilitated a systematic review of the wildfire prediction domain through an in-depth visual analytic process. The centrality of nodes within the network can indicate their importance, while the network density reflects the overall connectivity of the analyzed content.

### 2.4. Network Analysis and Visualization

Upon importing the data into CiteSpace, we conducted text mining and visual analysis [21] to identify the most cited articles, journals, and authors in the field of wildfire prediction. Additionally, we employed co-occurrence analysis [22] to identify the most common keywords and terms used in the publications. Furthermore, cluster analysis was applied to categorize papers in this field into distinct research areas.

To visualize the results, we utilized CiteSpace interactive network diagrams [23]. These diagrams assist in identifying key clusters and their interconnections. In the network diagrams, nodes (representing papers, authors, journals, or keywords) are depicted as circles, with the size of the circle indicating the number of occurrences in the dataset. The links between nodes represent co-citation or co-occurrence relationships, with the thickness of the link reflecting the strength of the relationship. CiteSpace's layout algorithms were utilized to arrange the nodes and links in a meaningful manner that emphasized key patterns and trends within the data.

## 3. Results

### 3.1. General Information

To provide a comprehensive overview of publications related to wildfire prediction, we determined and analyzed the number of publications per year employing the Web of Science and Scopus databases, as illustrated in Figure 1. Overall, there has been a noticeable increase in the number of publications in the field of wildfire prediction over the past nearly 50 years. From 1974 to 1990, the number of related publications was relatively low, with only seven papers published during this 16-year period. However, from 1991 to 2005, there was a gradual rise in the number of papers focused on forest fire prediction. Starting in 2006, research in the field entered a phase of rapid growth. The publication trends can be categorized into three distinct periods: the exploratory period (1974–1990), the period of steady growth (1991–2005), and the period of rapid development (2006–2022). Among these periods, the timeframe of 2016–2022 stands out as the most representative, demonstrating the highest growth rate for publications and accounting for approximately 58.6% of the total number of publications in the last 48 years. In summary, wildfire prediction has emerged as an increasingly active area of research, garnering significant attention from the scientific community.

Furthermore, to compare the two databases, Web of Science and Scopus, we have noted the number of articles they include for each year in Figure 1. Additionally, in Figure 2, we present the respective percentages of articles from each database. According to the data, Web of Science accounts for 80.6% of the articles, Scopus accounts for 24.5%, and there is an overlap of 5.1% where articles are indexed in both databases.

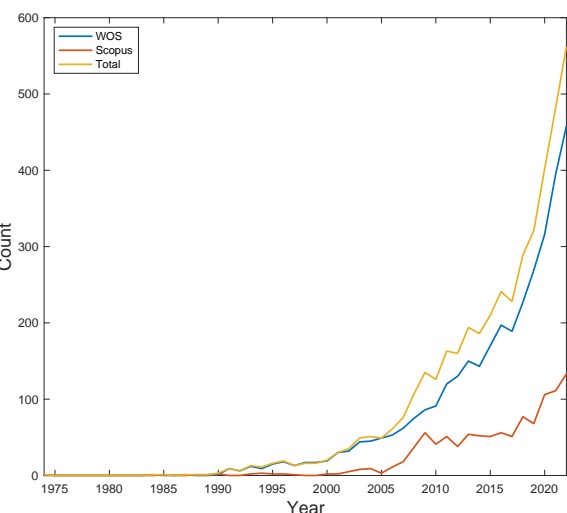

**Figure 1.** Quantity of wildfire prediction research in Web of Science and Scopus from 1974 to 2022.

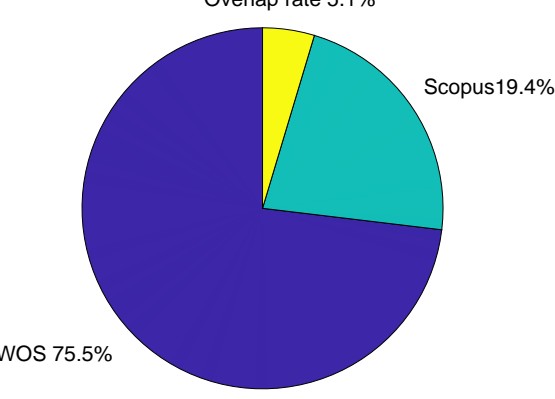

**Figure 2.** Percentagesof wildfire prediction studies in Web of Science and Scopus.

*3.2. Journal Distribution*

In our analysis of 4305 relevant publications, we examined the distribution of papers across different journals in the field of wildfire prediction. The top 20 journals each published no fewer than 40 relevant articles. The *International Journal of Wildland Fire* ranked first with 298 articles, followed by *Forest Ecology and Management* with 247 articles and *Remote Sensing* with 120 articles. Utilizing Bradford's law [24], we calculated the number of core journals in the field ($R_0$) using the formula $R_0 = 2\ln(e^E \times Y)$, where $E$ represents Euler's constant ($E \approx 0.5772$) and $Y$ represents the number of publications in the journal with the highest number of articles. By calculating $R_0$, we obtained a value of approximately 12.47. This estimation suggests that there are around 12 core journals in the field of wildfire prediction, which account for a substantial portion of the literature. These core journals serve as important outlets for researchers to disseminate their work and contribute to the advancement of knowledge in the field.

Knowledge of the number of core journals enables researchers to focus on key publications and stay updated with the latest developments in the field. Therefore, we present the 12 core journals in Table 1 and illustrate the proportion of papers in each journal in Figure 3. Based on Table 1 and Figure 3, we can observe the relative distribution of papers across different journals. The *International Journal of Wildland Fire* dominates with the highest proportion, followed by *Forest Ecology and Management*. Other journals, such as *Remote Sensing*, *Science of the Total Environment*, and *Forests*, also demonstrate their research significance in related fields. These findings provide valuable insights into the prominence of each journal within the academic community and the impact of their research areas.

**Table 1.** Ranking for published papers (top 12).

| Journal | Web of Science | Scopus | Total |
|---|---|---|---|
| *International Journal of Wildland Fire* | 286 | 12 | 298 |
| *Forest Ecology and Management* | 195 | 52 | 247 |
| *Remote Sensing* | 120 | 20 | 140 |
| *Science of the Total Environment* | 94 | 43 | 137 |
| *Forests* | 106 | 28 | 134 |
| *Atmospheric Chemistry and Physics* | 84 | 2 | 86 |
| *Atmospheric Environment* | 65 | 19 | 84 |
| *Journal of Geophysical Research Atmospheres* | 71 | 9 | 80 |
| *Atmosphere* | 61 | 18 | 79 |
| *Lecture Notes in Computer Science* | 58 | 18 | 76 |
| *Canadian Journal of Forest Research* | 60 | 16 | 76 |
| *Fire Switzerland* | 56 | 0 | 56 |

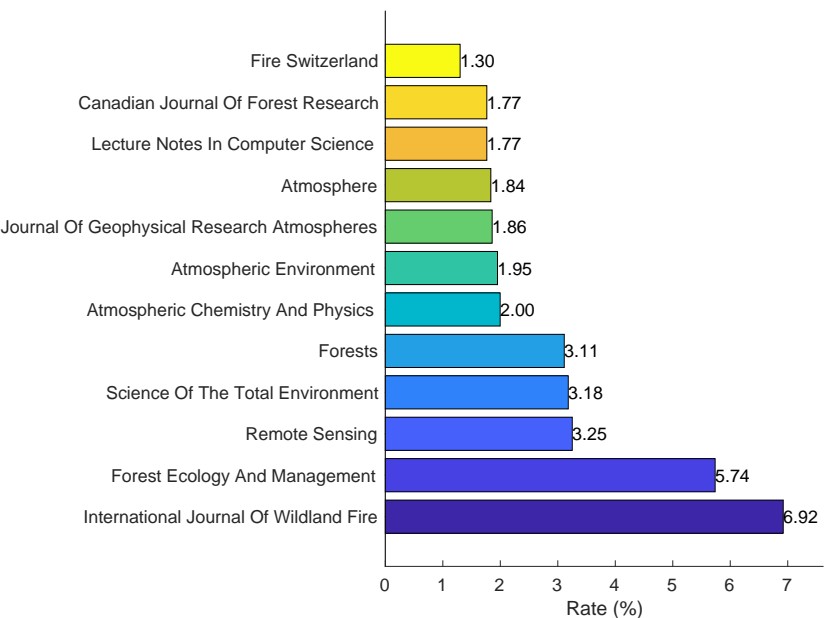

**Figure 3.** Percentages for total numbers of papers in the top 12 journals.

### 3.3. Network Analysis

#### 3.3.1. Author Network Map

The author co-occurrence knowledge map provides a visual representation of authors' influence and the extent of their collaborative relationships. To construct the co-author network, we utilized CiteSpace, setting the node type to Author, the time span to 1974–2022, and the time slice to 1 year. Running CiteSpace generated an author co-occurrence knowledge map with 1130 nodes, 1464 connected lines, and a density of 0.0023 (as depicted in Figure 4). At the top of Figure 4 is a colored band that corresponds to each year from 1974 to 2022 from left to right.

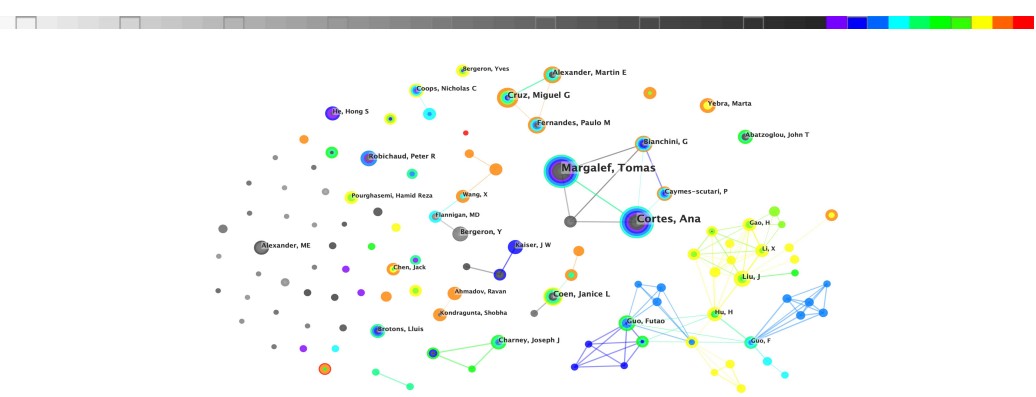

**Figure 4.** Coreauthor network map for wildfire prediction researchers.

In the author network map, each node represents an author, and the font size of the node labels corresponds to their publishing frequency. The links between nodes, displayed in different colors, indicate collaborations that have occurred at different times. The color spectrum ranges from gray to red, indicating earlier to more recent collaborations. To enhance the readability and aesthetics of the graph, authors with fewer than five publications are not shown by name due to the large number of nodes.

As can be observed in Figure 4, certain scholars have demonstrated continuous research contributions in the field of wildfire prediction over an extended period spanning

from 1974 to 2022. Notably, scholars such as Margalef Tomas and Cortes Ana exhibit multiple node colors, indicating their sustained engagement in research. Margalef has published 51 articles, while Cortes have published 47. Conversely, some scholars exhibit less continuity in their research, as evident from the presence of gray and blue nodes, which signify lower numbers of publications. This suggests that research outcomes necessitate continuous investigation and dissemination. To provide further insights, Table 2 presents statistics on authors with 10 or more publications.

**Table 2.** Authors with 10 or more articles.

| Rank | Author | Count |
| --- | --- | --- |
| 1 | Margalef Tomas | 51 |
| 2 | Cortes Ana | 47 |
| 3 | Miguel G Cruz | 17 |
| 4 | Janice L Coen | 15 |
| 5 | German Bianchini | 12 |
| 6 | Paulo M. Fernandes | 12 |
| 7 | Martin E. Alexander | 12 |
| 8 | Jianjun Liu | 11 |
| 9 | Yves Bergeron | 11 |
| 10 | Peter R. Robichaud | 10 |
| 11 | Futao Gao | 10 |

By combining Figure 4 and Table 2, we can observe that, during the period from 1974 to 2022, some authors engaged in close academic exchanges and collaborations with their peers. Notably, authors such as Futao Gao, Jianjun Liu, and Cortes Ana formed a relatively large network of cooperative relationships. Cortes Ana published related research in this field in 2006, along with German Bianchini. Their studies involved the utilization of computer models and algorithms for wildfire prediction, and their collaborative efforts influenced other researchers to participate in cooperative endeavors. Some authors demonstrated small-scale collaborations, such as Ravan Ahmadov, M.D. Flannigan, and Miguel G. Cruz. On the other hand, certain authors pursued independent research without any evident collaborative relationships, including Peter R. Robichaud, Marta Yebra, Lluis Brotons, and others.

Furthermore, during the 1970s and the end of the 20th century, most scholars exhibited limited collaboration, resulting in a relatively sparse author cooperation network, as indicated by the gray and black nodes in Figure 4. However, with the beginning of the 21st century, the frequency of cooperation and exchange among scholars significantly increased, as depicted by the blue, green, and orange nodes in Figure 4. This observed surge in collaboration can be attributed to the development of the social economy and the widespread adoption of the Internet and information technology.

### 3.3.2. Institution Network Map

We modified the node types in CiteSpace to generate an institutional co-occurrence knowledge graph, which consisted of 598 nodes, 3555 connections, and a density of 0.0199. To enhance the readability of the map, we utilized the Pathfinder Network method, retaining significant connections within the network. We also hid the names of institutions that published fewer than 70 papers, resulting in the creation of Figure 5. At the top of Figure 5 is a colored band that corresponds to each year from 1974 to 2022 from left to right.

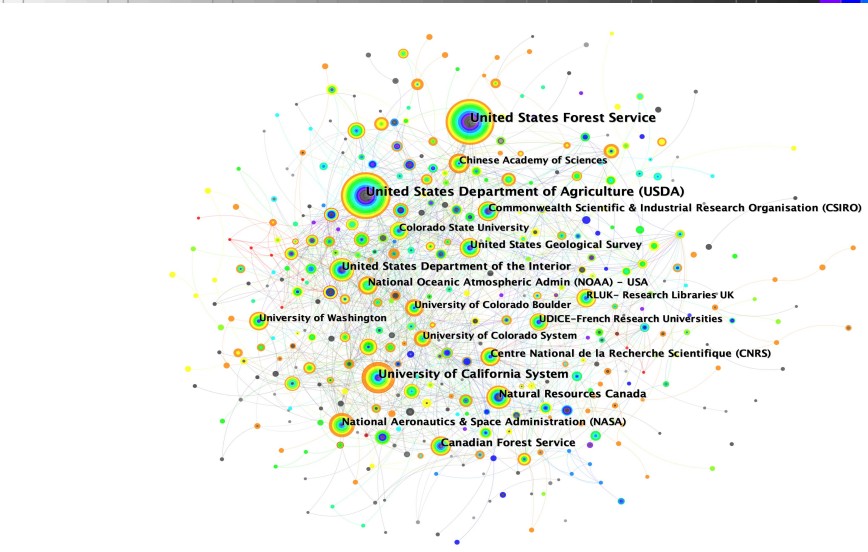

**Figure 5.** Institution network map for wildfire prediction research.

In Figure 5, each node represents an institution, with larger nodes indicating higher publication frequency. The colors of the nodes represent the publication times of the papers, with the range from grey to red indicating earlier to more recent publications. Based on Figure 5, we compiled a list of institutions that have published more than 100 papers, as shown in Table 3.

**Table 3.** Institutions with 100 or more papers.

| Rank | Institution | Count |
|------|-------------|-------|
| 1 | United States Department of Agriculture (USDA) | 402 |
| 2 | United States Forest Service | 376 |
| 3 | University of California System | 206 |
| 4 | Natural Resources Canada | 136 |
| 5 | United States Department of the Interior | 123 |
| 6 | Canadian Forest Service | 120 |
| 7 | National Aeronautics and Space Administration (NASA) | 113 |
| 8 | Commonwealth Scientific and Industrial Research Organisation (CSIRO) | 109 |
| 9 | United States Geological Survey | 105 |

By combining Table 3 and Figure 5, it becomes evident that research institutions involved in forest fire prediction predominantly consist of scientific research institutions and universities. Among them, the United States Department of Agriculture (USDA) has the largest node and is the institution with the highest number of published papers, totaling 402 articles. The United States Forest Service follows closely behind with 376 papers, while the University of California System ranks third with 206 articles. These findings indicate that these three institutions exhibit prominent research strengths among the various institutions involved in this field.

In terms of inter-institutional collaborations, numerous connections can be observed among different research institutions, forming distinct clusters. This suggests a strong sense of cooperation among institutions, with a high level of research outcome mobility.

For instance, the United States Geological Survey shows collaborative relationships with institutions such as Colorado State University, the United States Department of Agriculture (USDA), the University of Colorado System, and the University of California System, among others. Analyzing the node colors associated with each institution, it is evident that most institutions began publishing papers in this field after the year 2000. This indicates that research institutions have increasingly focused on forest fires and related topics since the beginning of the 21st century.

### 3.3.3. Country Network Map

Initially, we standardized the country names in the dataset by merging similar entries (e.g., "PEOPLES R CHINA", "TAIWAN", and "HONG KONG" were merged into "CHINA"; "UNITED STATES" into "USA"; etc.). Subsequently, by changing the node types in CiteSpace based on the literature data from Web of Science and Scopus, we generated a national co-occurrence knowledge map. The map had 128 nodes, 880 links, and a density of 0.1083, as illustrated in Figure 6. At the top of Figure 6 is a colored band that corresponds from left to right to each year from 1974 to 2022.

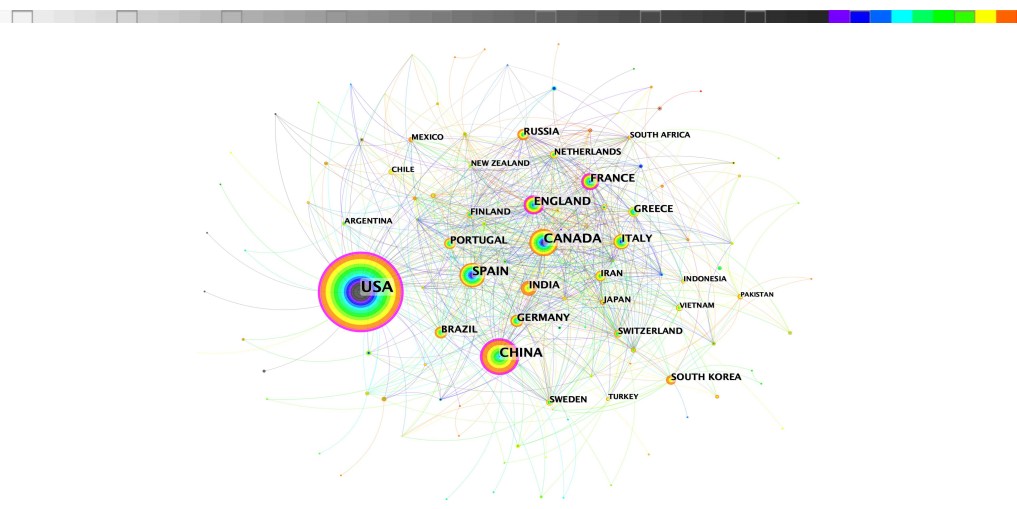

**Figure 6.** Country network map for wildfire prediction research.

Additionally, we compiled a list of the top 10 influential countries in this field, which is presented in Table 4. In Figure 6, each node represents a country, with larger nodes indicating a higher publication frequency. The colors of the nodes correspond to the publication times of the papers, with the range from grey to red indicating earlier to more recent publications.

**Table 4.** Top 100 countries with over 100 articles.

| Rank | Country | Count | Centrality |
|------|---------|-------|------------|
| 1 | USA | 2252 | 0.44 |
| 2 | China | 474 | 0.26 |
| 3 | Canada | 468 | 0.06 |
| 4 | Spain | 356 | 0.09 |
| 5 | UK | 201 | 0.12 |
| 6 | France | 178 | 0.11 |
| 7 | Italy | 138 | 0.09 |
| 8 | Portugal | 132 | 0.02 |
| 9 | Germany | 129 | 0.06 |
| 10 | India | 117 | 0.05 |

As seen in Figure 6 and Table 4, the country distribution reveals notable trends. The USA, China, the UK, and Spain exhibit high centrality in the network structure, indicating their significant research contributions in the field from 1974 to 2022, with each country publishing over 200 papers. The USA, in particular, ranks first with a centrality of 0.44, highlighting its strong overall research prowess in this field. The data demonstrate that the USA published 2252 articles during this period.

Regarding international collaborations, scholars from various countries have engaged in frequent cooperation in this research field. The data reveal that the USA, ranking first in centrality, has had collaborative relationships with 69 countries. Similarly, China, ranking second, has had collaborative relationships with 42 countries in this research field.

### *3.4. Keywords Analysis*

3.4.1. Co-Occurrence Analysis Results

Keywords represent the condensed core research content of an article and reflect its thematic focus. Analyzing keywords can help uncover research hotspots and subject content within a field. The keyword spatiotemporal zoning map illustrates the evolution of research hotspots over time. By utilizing keywords as nodes in CiteSpace, we generated a keyword co-occurrence knowledge graph with 1101 nodes, 9137 connections, and a density of 0.0151. Using the Pathfinder Network method and retaining important connections within the network, we obtained Figure 7. We have plotted a colored band at the top of Figure 7, which corresponds to each year from 1974 to 2022 from left to right. In Figure 7, each node represents a keyword, with larger nodes indicating more frequent occurrence of the keyword. The colors of the nodes correspond to the publication times of the papers, with the range from grey to red indicating earlier to more recent publications.

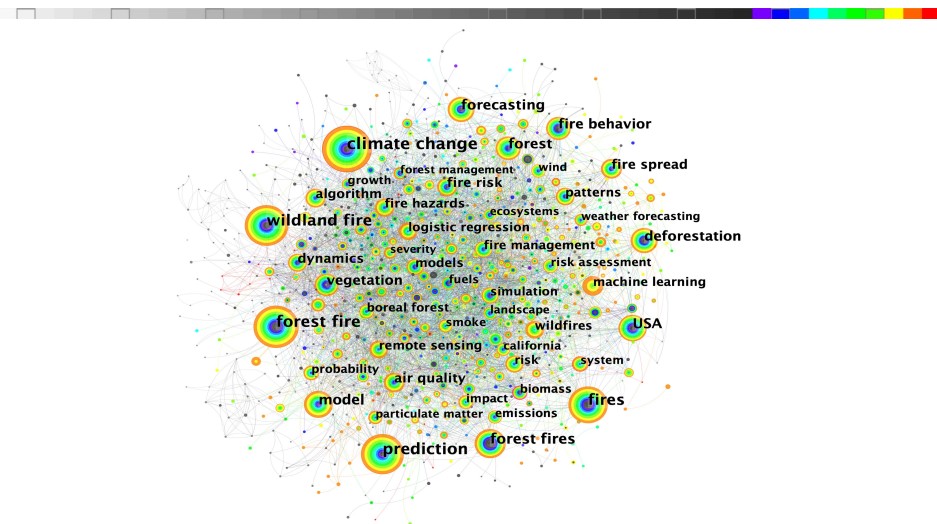

**Figure 7.** Keyword network map for wildfire prediction research.

Among the keywords in the co-occurrence network, the top 10 most frequently associated with wildfire prediction were "climate change", "forest fire", "prediction", "wildland fire", "fires", "model", "forest", "USA", "forecasting", and "deforestation" (as shown in Table 5). In the network, the size of each node corresponds to the frequency of occurrence of the respective keyword, while the color of the node indicates the publication time, with darker colors representing earlier release dates.

**Table 5.** Top 10 high-frequency keywords.

| Rank | Keyword | Count | Centrality |
|------|---------|-------|------------|
| 1 | Biomass | 0.12 | 136 |
| 2 | Forestry | 0.09 | 198 |
| 3 | Ecosystems | 0.06 | 108 |
| 4 | Canada | 0.06 | 90 |
| 5 | Model | 0.05 | 443 |
| 6 | USA | 0.05 | 380 |
| 7 | Dynamics | 0.05 | 253 |
| 8 | Air quality | 0.05 | 253 |
| 9 | Boreal forest | 0.05 | 156 |
| 10 | Weather forecasting | 0.05 | 113 |

Table 5 presents the top 10 keywords related to wildfire prediction along with their rankings, frequencies, and centrality in the co-occurrence network. The keyword "climate change" ranked first with a frequency of 1119, indicating its significance in understanding the impact of environmental shifts on wildfire occurrences. Following closely was "forest fire" with a frequency of 951, emphasizing the relevance of studying fire incidents in forested areas. "Prediction" ranked third with 861 occurrences, highlighting its importance in forecasting future wildfire events. The keyword "wildland fire" appeared 810 times, underscoring the need for comprehensive research on fires in natural landscapes. Interestingly, the keyword "fires" had a relatively high frequency of 699 and stood out with a centrality value of 0.09, suggesting its influential role in the co-occurrence network. The keyword "model" appeared 443 times, indicating its significant presence in wildfire prediction studies. Other keywords in the top 10 included "forest" (393 occurrences), "USA" (380 occurrences), "forecasting" (368 occurrences), and "deforestation" (354 occurrences), each contributing valuable insights to the field of wildfire prediction research.

Overall, from 1974 to 2022, research in the field of wildfire prediction has focused on several key areas. Firstly, there has been a strong emphasis on utilizing computer models and algorithms, such as the random forest algorithm, machine learning, and logistic regression, to enhance predictive capabilities and understand the complex dynamics of wildfires. These modeling techniques have played a crucial role in improving forecasting accuracy and aiding in fire management strategies. Secondly, researchers have devoted significant attention to investigating the causes of fires, exploring the interplay between subjective factors, such as climate change, and objective factors, such as human activities, that contribute to fire occurrences. Factors such as climate change, temperature, fuel availability, weather patterns, and drought have been extensively examined to comprehend their influence on fire behavior and occurrence patterns. Additionally, research efforts have been dedicated to assessing the risks and hazards posed by fires, including evaluating fire severity, impact assessments, and studying the implications of fires for air quality. Furthermore, there has been a notable focus on studying ecosystems in the aftermath of fires, landscape changes, vegetation dynamics, and post-fire recovery processes. These research themes have collectively contributed to a comprehensive understanding of wildfires and play a vital role in informing fire management and mitigation strategies.

Using the analysis of the keyword co-occurrence network, the main research directions and hot issues in this field can be identified, providing important references for researchers. In addition, as another important result of the keyword co-occurrence analysis, the indicator of centrality can be regarded as highlighting the key, turning, and triggering points of the research field. We have listed the top 10 keywords with high centrality in Table 6.

**Table 6.** Top 10 keywords in terms of centrality.

| Rank | Keyword | Centrality | Count |
|------|---------|-----------|-------|
| 1 | Biomass | 0.12 | 136 |
| 2 | Forestry | 0.09 | 198 |
| 3 | Ecosystems | 0.06 | 108 |
| 4 | Canada | 0.06 | 90 |
| 5 | Model | 0.05 | 443 |
| 6 | USA | 0.05 | 380 |
| 7 | Dynamics | 0.05 | 253 |
| 8 | Air quality | 0.05 | 253 |
| 9 | Boreal forest | 0.05 | 156 |
| 10 | Weather forecasting | 0.05 | 113 |

The top 10 keywords with high centrality in the co-occurrence network were "biomass", "forestry", "ecosystems", "Canada", "model", "USA", "dynamics", "air quality", "boreal forest", and "weather forecasting". Centrality measures the importance of a node in connecting different parts of the network, and the higher the value, the more critical the node is in the network. The high centrality of these keywords indicates that they play crucial roles in connecting various aspects of the research field and have a significant influence on the overall network structure. For example, keywords such as "biomass", "forestry", and "ecosystems" suggest the importance of studying the impact of these factors on the research domain. Similarly, keywords such as "Canada" and "USA" indicate the significance of the research conducted in these countries, possibly reflecting their prominent contributions and expertise in the field. The presence of keywords such as "model", "dynamics", and "weather forecasting" suggests a focus on predictive modeling and understanding the dynamic nature of the research field. Overall, these keywords with high centrality highlight important themes and areas of interest within the research network.

3.4.2. Keyword Cluster Analysis

After clustering analysis, 43 clusters were obtained, and the modularity Q = 0.4729 and the silhouette S = 0.7326, indicating that the clustering of keywords was reasonable and highly reliable. The size of each cluster is indicated by the label number of the nodes, with labels with lower numbers representing larger cluster sizes. The clustering results are shown in Figure 8. In Figure 8, nodes represent keywords, and keywords belonging to the same cluster are divided into the same region. The node color and region color are consistent with the label color of the cluster they belong to.

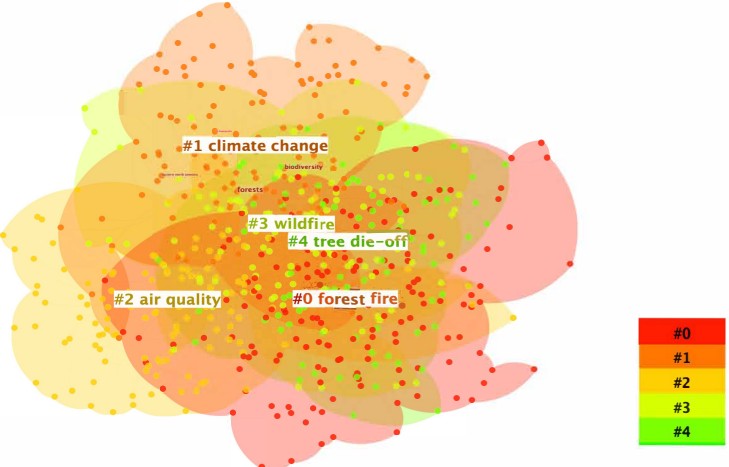

**Figure 8.** Keywordcluster analysis for wildfire prediction research.

According to the clustering labels, the largest cluster was "forest fire", which primarily encompassed research on fire-related issues using computer models and algorithmic

approaches. This cluster included research investigating the effects of vegetation types and emissions on fire spread. It included keywords such as "machine learning", "neural network", "vegetation types", and "biomass burning emissions".

The second cluster was "climate change" and included research that explored the impact of fires on animals and plants in natural ecosystems, focusing on the changes brought about by fire incidents. Keywords in this cluster included "biodiversity", "tree species performance", and "surface".

The third cluster was "air quality" and included research that utilized factual measurement data to study increases in particulate matter in the air caused by fires, air pollution issues, and the presence of particulate matter in the air after fires. Keywords in this cluster included "black carbon", "air pollution", "smoke", and "biomass burning".

The fourth cluster was "wildfire" and included research that examined the responses of relevant institutions and management departments after fires. It encompassed research on decision-making processes, fire management strategies, and risk assessment. Keywords in this cluster included "decision making", "fire management", and "risk assessment".

The fifth cluster was "tree die-off" and included research that investigated the reasons behind declines in vegetation rates and reductions in vegetation following fires. This research examined dynamic changes in data and explored topics such as soil erosion, watershed management, and debris flow. Keywords in this cluster includex "soil erosion", "watershed", and "debris flow".

### 3.4.3. Timeline Analysis

The cluster mapping of keywords was transformed into a timeline diagram, as shown in Figure 9. This diagram provides a clear visualization of the evolution of forest wildfire research over time. In the diagram, the size of the nodes corresponds to the frequency of occurrence, while the horizontal axis represents the timeline. The nodes are arranged from left to right in the order in which the corresponding keywords appeared over time. The labels for the different clustering categories are displayed on the right-hand side of the graph.

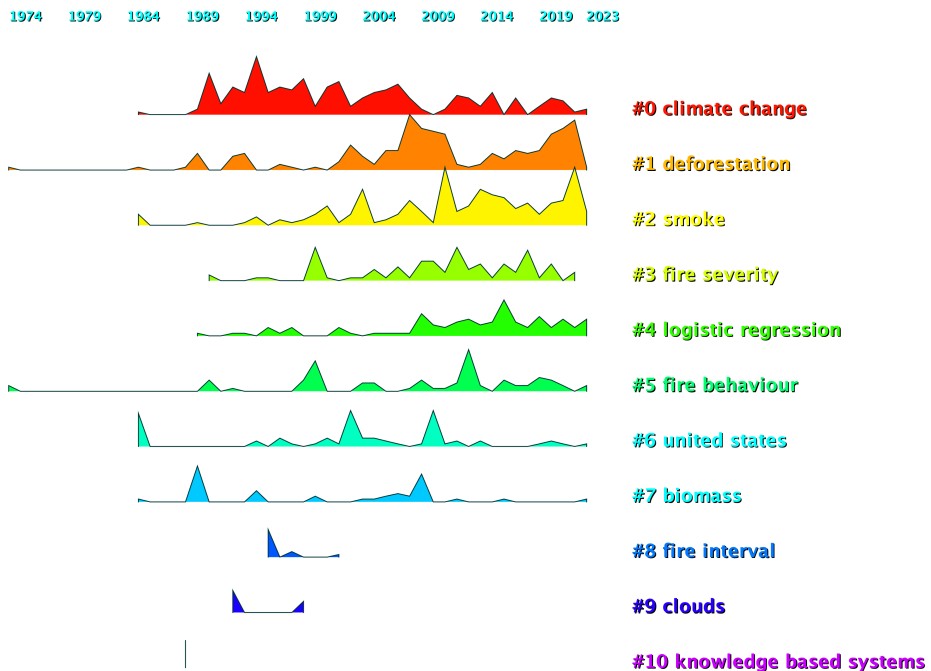

**Figure 9.** Landscape map of research hotspots for wildfire prediction.

From the analysis diagram in Figure 9, we can observe that, since 1984, researchers have been actively engaged in studying topics related to "climate change", "smoke",

"biomass", and other factors. This indicates that research focusing on factors such as climate change, smog, and biomass has gradually gained prominence in wildfire prediction studies, particularly after 1989. Moreover, research on "deforestation" and "fire behavior" also started earlier and garnered attention from researchers after 1974. From the timeline analysis, we can also identify the emergence of other keywords. For instance, "fire severity" and "fire interval" may represent more detailed examination of the severity of and time intervals between wildfire occurrences. Additionally, keywords such as "logistic regression" and "knowledge-based system" suggest the existence of research utilizing statistical analysis methods and knowledge-based systems to predict wildfires.

In summary, the timeline analysis presented in Figure 9 illustrates the development trends and shifts in focus within wildfire prediction research across different topics. These keywords provide a comprehensive perspective and contribute to our understanding of the evolution of the research field and the inter-relationships between various topics.

### 3.4.4. Burst Term Analysis

Burst term analysis can detect high-frequency keywords that appear in research over a period of time, helping us to understand the frontiers of research, shifts in research focus, and the latest research developments. In Figure 10, "Year" indicates the year when the keyword first appeared, "Begin" and "End" represent the starting and ending years of the keyword as a frontier, while "Strength" indicates the emergence intensity, and the red bar reflects the specific time period when the keyword became a research hotspot. As shown in Figure 10, the top five keywords with the strongest emergence intensity were "conservation", "North America", "mathematical model", "regression analysis", and "regimes". These emerging keywords indicate that there is a growing emphasis on topics such as conservation efforts, particularly in the context of North America. The inclusion of "mathematical model" and "regression analysis" suggests an increased focus on quantitative methods and data analysis in the research. Additionally, the presence of "regimes" implies a focus on understanding different patterns or states within the research domain.

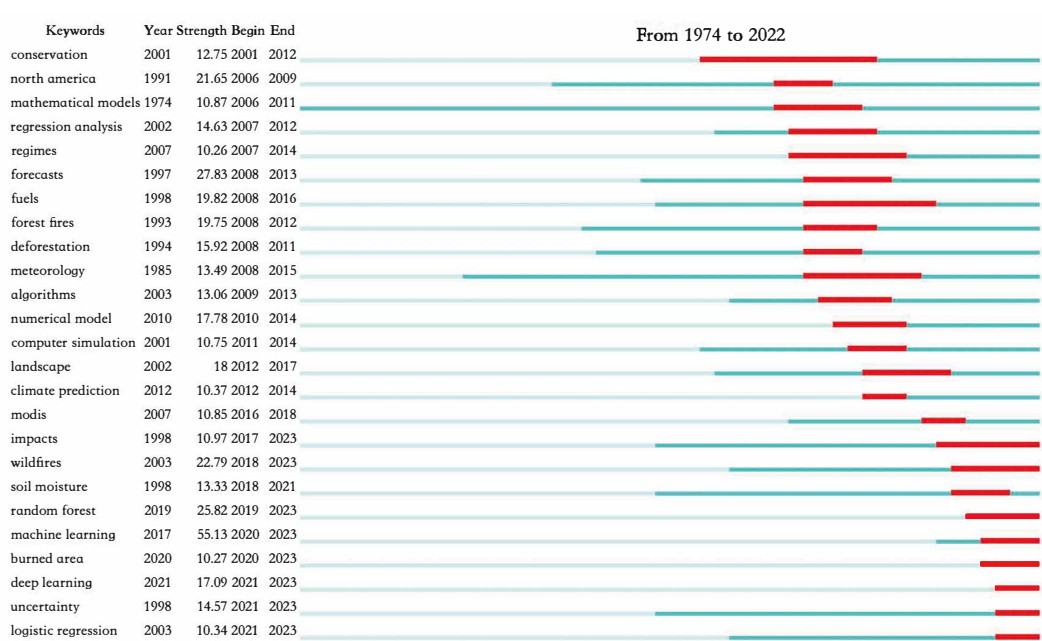

**Figure 10.** Top 25 Keywords with the Strongest Citation Bursts

The emergence of these keywords reflects the shifting interests and priorities within the field, which demonstrate a particular emphasis on conservation, regional contexts, and quantitative approaches. These trends highlight the importance of studying and addressing conservation challenges in North America, as well as the need for rigorous mathematical modeling and data analysis techniques to enhance understanding and prediction in the

field. The identification of emerging keywords can serve as a guide for researchers and practitioners, helping them stay informed about the latest developments and contributing to the advancement of knowledge in the field of wildfire prediction.

Overall, the analysis of emerging keywords and their trends provides valuable insights into the current research landscape and future directions in the field of wildfire prediction. The timeline analysis highlights the evolution of research themes over time, with certain topics gaining prominence at different stages. This indicates the dynamic nature of the field and the evolving interests of researchers.

## 4. Discussion

The analysis conducted in this study revealed several key research themes in the field of wildfire prediction. These themes included the use of computer models and algorithms, the study of climate change and its impact on wildfires, the assessment of fire risks and hazards, the investigation of post-fire ecosystems and recovery processes, and the examination of factors such as deforestation and fire behavior. These themes reflect the multidimensional nature of wildfire prediction research and highlight the importance of understanding various aspects related to fire occurrence, spread, and management.

The analysis of author networks identified several influential authors in the field of wildfire prediction. These authors have made significant contributions to the research through their extensive publication records and collaborative relationships. Their research expertise and knowledge have had a substantial impact, helping to shape the field and advance our understanding of wildfire prediction. Collaborative efforts among these influential authors have also played a crucial role in promoting knowledge exchange and fostering research collaborations.

The timeline analysis showcased the evolution of the field of wildfire prediction in recent decades. It revealed distinct periods of research development, from an exploratory phase to steady growth and rapid development. The increasing number of publications and the emergence of new research topics reflect the growing importance of wildfire prediction in addressing the challenges posed by wildfires. The timeline analysis also highlighted the changing research priorities and the dynamic nature of the field, as new topics and approaches have gained prominence over time.

The analysis of keyword networks identified research hotspots and potential gaps in the field of wildfire prediction. Hotspots such as climate change, fire behavior, and predictive modeling indicate active areas of research and ongoing advancements. However, the identification of emerging keywords also suggested potential research gaps that require further exploration. These gaps may include understudied topics, limited geographical coverage, or areas where more in-depth analysis and investigation are needed. Recognizing these hotspots and gaps can guide future research efforts and help address the existing knowledge limitations.

While this study provides valuable insights into the field of wildfire prediction, certain limitations must be acknowledged. The analysis was conducted based on a specific set of databases—namely, Web of Science and Scopus—which may not cover all relevant publications in the field. In addition, the analysis focused on specific criteria for the selection of articles, which may have excluded certain types of publications or areas of research. It is crucial to consider these limitations when interpreting the findings and to complement the analysis with additional sources of information and research data. In addition, the focus of this study on English-language publications and the exclusion of relevant studies in other languages may have limited the scope and representativeness of the findings.

Based on the findings of this study, several potential future research directions can be identified. Firstly, there is a need for continued research on the development and refinement of predictive models, incorporating new technologies and data sources to enhance accuracy and efficiency. Secondly, further investigation into the impacts of climate change on wildfire occurrences and behavior is necessary, considering the changing environmental

conditions. Thirdly, studies focusing on post-fire ecosystems, including vegetation recovery and ecological dynamics, can contribute to better understanding and management of fire-affected areas. Lastly, interdisciplinary collaborations and knowledge exchange between researchers, policymakers, and practitioners can help bridge the gap between research findings and practical applications, leading to more effective wildfire prediction and mitigation strategies.

## 5. Conclusions

In conclusion, this comprehensive bibliometric analysis sheds light on the research landscape of wildfire prediction. The study revealed the growth and development of the field, identified key research themes and gaps, and highlighted influential authors and institutions. The findings provide valuable insights for researchers, policymakers, and practitioners in understanding the trends, challenges, and opportunities in wildfire prediction.

The analysis underscored the importance of interdisciplinary collaboration and the integration of advanced technologies, such as computer models and algorithms, remote sensing, and artificial intelligence. It emphasized the significance of addressing climate change impacts, studying fire behavior, assessing risks and hazards, and understanding post-fire ecosystem dynamics. These areas of focus can contribute to improved wildfire forecasting and effective decision making.

The study also emphasized the need for standardized methods and data formats to enhance comparability and reproducibility in wildfire prediction research. It highlighted the potential of integrating diverse data sources and leveraging emerging techniques, such as machine learning and spatial analysis. Furthermore, the analysis identified research gaps that require further attention, such as the regarding complexity of wildfires, the lack of comprehensive prediction systems, and the need for real-time monitoring and early warning systems.

By considering the insights and recommendations from this study, researchers and practitioners can shape their future research agendas and strategies. Collaboration across disciplines and institutions, along with the adoption of innovative approaches and technologies, can propel the field of wildfire prediction forward. Ultimately, advancing wildfire prediction capabilities will contribute to effective wildfire management, risk reduction, and the protection of human lives and ecosystems.

**Author Contributions:** M.P. and S.Z. conceptualized the study and wrote the original draft. M.P. performed the numerical simulations. S.Z. was responsible for data curation and reviewed and edited the manuscript. All authors have read and agreed to the published version of the manuscript.

**Funding:** This research received no external funding.

**Institutional Review Board Statement:** Not applicable.

**Informed Consent Statement:** Not applicable.

**Data Availability Statement:** Not applicable.

**Conflicts of Interest:** The authors declare no conflict of interest.

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
