# Peer review of "Visualization of Prediction Methods for Wildfire Modeling Using CiteSpace: A Bibliometric Analysis"

_atmosphere, doi:10.3390/atmos14061009_

Round 1

Reviewer 1 Report

This paper provides an overview of the current trends and research gaps in 2 wildfire prediction by con- ducting a bibliometric analysis of papers in the Web of Science database. The following comments should be addressed.

1. The databasej is limited to Web of Science. Is this limitation reasonable? The database should be as broad as possible. The references are very limited.

2. The quality of figures are too low. These must be improved.

It is ok.

Author Response

In response to your suggestion, we have responded in the attached document.

Reviewer 2 Report

The report gives recommendations for future research, such as adding new data sources and applying cutting-edge approaches, and helps to better understand research trends and gaps in wildfire prediction. The authors are expected to consider following points during the revision:

1. Organization of the article needs to be added at the end of introduction section.

2.  The research gaps and contributions should be listed. 

3.  The authors considered the analysis only for 22 years. If the number of years increased, the outcome may be better.

4. There are many databases other that Web of Science like Scopus, Semantic Scholar, Google Scholar, ScienceOpen. The authors should consider these databases also in future for better outcome. 

5. This approach needs to be compared with similar approaches for validation. 

Minor tuning/corrections required. 

Author Response

(The authors gave the same response as above.)
